# Anxiety and Depression and Health-Related Quality of Life in Adults with Gastroesophageal Reflux Disease: A Population-Based Study

**DOI:** 10.3390/healthcare11192637

**Published:** 2023-09-27

**Authors:** Monira Alwhaibi

**Affiliations:** 1Department of Clinical Pharmacy, College of Pharmacy, King Saud University, Riyadh 11437, Saudi Arabia; malwhaibi@ksu.edu.sa; Tel.: +966-535-384-152; 2Medication Safety Research Chair, College of Pharmacy, King Saud University, Riyadh 11437, Saudi Arabia

**Keywords:** anxiety, depression, gastroesophageal reflux disease, quality of life, SF-12

## Abstract

Background: Adults with gastroesophageal reflux disease (GERD) are susceptible to mental disorders that might significantly affect their health-related quality of life (HRQoL). Therefore, the purpose of this study was to evaluate how HRQoL in individuals with GERD is related to coexisting anxiety and depression. Methods: A cross sectional study was conducted among 3068 adult patients 22 years of age and older with GERD diagnoses have been identified using the data from the Medical Expenditure Panel Survey from 2017 to 2020 for United States adults. Data are gathered by MEPS using an overlapping panel design over a period of two and a half years. HRQoL was established using the Physical and Mental Component Summary (PCS & MCS) values from the SF-12. Multivariate forward linear regression analysis was used to assess the relationship between anxiety, depression, and HRQoL in people with GERD after accounting for various factors. Results: Of the 3068 people with GERD who had been identified, 56.4% were women, 59.4% were in their 50s or 60s and 64.8% were employed. Thirteen per cent of people with GERD had depression, thirteen per cent had anxiety, and ten per cent had both. Adults with concurrent anxiety and depression had the lowest mean PCS and MCS scores compared to those with GERD. After adjusting for all independent factors, GERD patients with anxiety (MCS = −10.819, *p*-value < 0.0001), depression (MCS = −6.395, *p*-value < 0.0001), and both (MCS= −42.869, *p*-value < 0.0001) had substantially worse HRQoL than those without these comorbidities. Notably, better HRQoL scores were positively associated with marital status, employment, perceived overall health, and physical activity. Conclusions: The results from this nationally representative sample shed insight into the relationships between low HRQoL and anxiety and depression among individuals with GERD. It also demonstrated the detrimental impacts of co-occurring chronic illnesses, low socioeconomic status, and the positive benefits of employment and exercise on HRQoL. This study emphasizes the clinical, policy, and public health implications for better healthcare, allocation of resources, and promotion of lifestyle modifications to improve the HRQoL in patients with GERD.

## 1. Introduction

Gastroesophageal reflux disease (GERD) is a chronic, often recurring, persistent condition affecting the digestive system’s upper tract. GERD refers to a condition that develops when the reflux of stomach contents causes troublesome symptoms or complications [1]. Heartburn, chest discomfort, and regurgitation are some examples of typical GERD symptoms [2,3]. GERD is one of the most common medical conditions globally and has been recognized as a significant healthcare issue [4]. According to recent estimates, 13% of people worldwide have GERD, and its incidence is increasing [4]. The estimated worldwide burden of GERD in 2019 was 6.0 million years lived with disability (YLDs), 783 million prevalent cases, and 309 million incident cases [5].

GERD has been associated with various illnesses, such as cancer, psychological and cardiovascular problems [6,7,8]. It harms health, lowers productivity and everyday activities, and impairs health-related quality of life (HRQoL) [6,9,10,11,12]. GERD sufferers are more predisposed to experience significant anxiety and depression. According to a nationwide population-based cohort study of adults, those with GERD had a threefold increased risk of depression and anxiety disorder [7]. Comorbidity of depression or anxiety may impair GERD sufferers’ Health-related quality of life (HRQOL) and increase illness burden [13,14,15]. A post hoc examination of a prospective cohort of 98 patients with GERD at a tertiary referral hospital in Spain revealed an independent relationship between the SF-36 mental component and state anxiety and depression [15]. A study in China among 279 patients with GERD examined the psychological aspects of gastroesophageal reflux disease (GERD) and how they affected patients’ health-related quality of life (HRQOL). It found that all dimensions of SF-36 that measure HRQoL were negatively correlated with anxiety and depression [14]. Anxiety and depression exhibited considerable negative associations with both the physical and mental health of HRQoL in GERD patients, according to another cross-sectional research of 358 GERD patients in Wuhan, China [16]. In a prospective study of 147 GERD patients in the Netherlands, Kessing et al. found that anxiety and depression levels were linked to lower scores for the mental component of HRQoL [17].

For GERD patients, health-related quality of life is a critical outcome determinant that refers to a person’s functions and their perceived physical, mental, and social well-being [18]. One of the key objectives for individuals with GERD is to keep a positive HRQoL despite the difficulties that come with their condition. Therefore, identifying the reasons for a lower HRQoL is essential, particularly in light of the comorbidities related to mental health. There are currently no population-based published data in the US that evaluate the psychological comorbidities of GERD among adults. Additionally, little research assesses the psychological comorbidities’ effects on GERD’s health-related quality of life globally. Previous studies utilized self-reported measures to evaluate anxiety and depression [15,16,17], whereas this study used clinical diagnosis codes. As a result, we investigated GERD in a representative adult population sample. We sought to ascertain the prevalence of psychological disorders among adults with GERD and the influence of psychological comorbidities on GERD health-related quality of life after adjusting for a wide variety of confounders, such as socioeconomic factors, access to treatment, physical activity, and medical comorbidities.

## 2. Materials and Methods

### 2.1. Study Design and Data

A cross-sectional study was conducted using data from the Medical Expenditure Panel Survey (MEPS) for 2017, 2018, 2019, and 2020. MEPS is a national survey that gathers information on sociodemographic characteristics, health insurance, medical issues, medication usage, and other health services from the non-institutionalized US civilian population. Data are gathered by MEPS using an overlapping panel design over a period of two and a half years, starting with a preliminary contact and continuing with five rounds of interviews.

### 2.2. Study Population

The study inclusion criteria are as follows: (1) adults between the ages of 22 and 64; (2) had been diagnosed with gastroesophageal reflux disease (GERD); (3) were alive during the above-mentioned calendar years; (4) and had no missing HRQoL data. Individuals with a GERD diagnosis were identified from the MEPS medical conditions file using the clinical diagnostic codes from the International Classification of Diseases, Ninth Revision, Clinical Modification (ICD-9-CM).

### 2.3. Measures

#### 2.3.1. Outcome: Health-Related Quality of Life (HRQoL)

To determine their HRQoL, adults who participated in the MEPS and were at least 18 years old had to self-administer questionnaires. Using the Physical Component Summary (PCS) and Mental Component Summary (MCS) of the Short-Form 12 Version 2 (SF-12V2) questionnaire, the physical and mental components of HRQoL were assessed [19]. Better physical and mental HRQoL is indicated by higher PCS and MCS scores, respectively. The MEPS’s Mental Component Summary Scores (MCS) and Physical Component Summary Scores (PCS) were both found to have high internal consistency reliability, adequate reliability, and validity, and should be appropriate for use in a variety of database-based projects in groups with and without cognitive limitations [19].

#### 2.3.2. Independent Variables

Adults with GERD were divided into four groups (GERD alone, GERD and anxiety, GERD and depression, and GERD and both disorders), each of which were mutually exclusive.

Sociodemographic parameters such as gender, age in years, race or ethnicity, marital status, income, area of residence, education level, health insurance, prescription insurance, job status, and poverty status, were included as independent variables. Poverty status was classified in four groups based on family income with respect to the federal poverty line (FPL): poor, near poor, middle-income, and high-income. Other factors were exercise, concomitant chronic conditions, and subjective physical health.

Using clinical classification codes and ICD-9 cm codes, concurrent chronic illnesses have been identified in the MEPS file. Following verification by expert coders, MEPS researchers converted diseases from household descriptions into clinical classification codes. Expert coders, for instance, used the ICD-9 cm codes 296, 300, and 311 for depression to report individuals with depression in the MEPS file.

### 2.4. Statistical Analyses

The study population was described using descriptive statistics. The characteristics of the GERD groups were compared using chi-square tests. Using ANOVA, meant that differences in HRQoL between GERD groups were identified. Multivariable forward linear regression model took into account all independent variables (such as sociodemographic factors, physical activity, health insurance coverage, and concurrent chronic health conditions) to assess the relationship between GERD groups and HRQoL. A *p*-value lower than 0.05 was deemed statistically significant. To account for all estimations, person-level weights and variance adjustment weights (strata and primary sample unit) from the MEPS were used in the statistical analyses. SAS 9.4 was used to analyze the data (SAS Institute Inc., Cary, NC, USA).

## 3. Results

### 3.1. Study Sample Characteristics

Table 1 displays the characteristics of the study sample; 3068 individuals with GERD were included in this study. Women (56.4%), adults in their 50s or 60s (59.4%), and those who were employed (64.8%) made up the majority of the sample of adults with GERD. Thirteen per cent of people with GERD had depression, thirteen per cent had anxiety, and ten per cent had both conditions.

Women with GERD experienced considerably higher rates of depression (15.9% vs. 9.9%) and anxiety (12.9% vs. 7.4%, *p*-value < 0.0001) than males did. Additionally, GERD patients who were unemployed had significantly higher rates of depression (17.7% vs. 10.9%) and comorbid anxiety and depression (15.4% vs. 7.8%, *p*-value < 0.0001) than those who were in employment. Additionally, patients with GERD who had these comorbidities (hypertension, diabetes, asthma, Chronic Obstructive Pulmonary Disease (COPD), arthritis, and osteoporosis) had a considerably higher percentage of depression and anxiety than adults with GERD without these comorbidities (*p*-value < 0.0001). 

### 3.2. Mean Health-Related Quality of Life Scores by GERD Groups

There were significant differences between the GERD groups in the PCS and MCS scores for HRQoL (Table 2). For instance, the mean PCS score was lower in GERD patients who also had depression and anxiety (Mean = 33.41) than it was in the other groups (GERD only, GERD and depression, and GERD and anxiety, which had mean PCS scores of 43.90, 38.67, and 39.86, respectively). Similarly, the MCS score was lower in GERD patients who also had depression and/or anxiety compared to other groups (35.61 for GERD and depression and/or anxiety, 39.90 for GERD only, and 38.12 for GERD & anxiety).

### 3.3. Multivariate Adjusted Analysis of Factors Associated with HRQoL in Patients with GERD

Table 3 shows how GERD groups and HRQoL are related after adjustment. After controlling for all independent variables, GERD patients with depression (MCS: = −6.395, *p*-value = 0.0001), anxiety (MCS: = −2.869, *p*-value = 0.0001), and both (MCS: = −10.819, *p*-value = 0.0001) had significantly poorer HRQoL than those without these comorbidities.

Factors negatively associated with HRQoL include young age, female gender, poverty status, health insurance, and comorbidities, including heart disease, hypertension, diabetes, hyperlipidemia, asthma, COPD, arthritis, and cancer. For instance, when compared to people with high income, poor adults had lower HRQoL for both their physical health summary score (PCS: = −2.230, *p*-value < 0.0001) and their mental health summary score (MCS: = −0.363, *p*-value < 0.0001).

Factors positively associated with HRQoL included marital status, employment, perceived general health, and physical activity. For instance, persons who were employed had greater HRQoL than those who were unemployed in terms of both the physical health summary score (PCS: = 5.909, *p*-value < 0.0001) and the mental health summary score (MCS: = 2.819, *p*-value < 0.0001).

## 4. Discussion

This research provides insight into the relationship between GERD sufferers’ physical and mental comorbidities and HRQoL. According to our study, patients with GERD and comorbidities of depression and anxiety had considerably lower health-related quality of life (HRQoL) than those without these comorbidities.

According to earlier published studies, in individuals with GERD, mental comorbidities have a negative impact on HRQoL. For example, studies in Spain [15], China [14,16], and the Netherlands [17], reported that all dimensions of SF-36 that measure HRQoL were negatively correlated with anxiety and depression in GERD patients. According to recent a systematic review and meta-analysis, GERD patients’ levels of anxiety and depression were higher than those of healthy controls [20]. Based on the pooled results of 30 studies, the prevalences of anxiety and depressive symptoms were 34.4% and 24.2% in GERD patients, respectively [20]. Additionally, a positive correlation between heartburn symptoms and psychological disorders was discovered [21].

Factors negatively associated with HRQoL in this study included young age, female gender, poverty status, health insurance, and concurrent comorbidities. With regard to gender and income, our finding is consistent with previous studies that reported an association between the female gender and lower physical component of the SF36 [15] and low income and poor HRQoL [22]. Maintaining people’s health is heavily influenced by income, a fundamental measure of social class. Higher income is associated with improved health and reduced health risks, whereas lower income is associated with greater exposure to risk factors for developing diseases [22]. In terms of health insurance, our findings shed light on the negative relationship between having health insurance and poor HRQoL, this finding contradicts the findings by Bharmal et al., who used MEPS data for the year 2000 and reported that individuals without health insurance had significantly lower mean PCS scores and MCS scores than those with health insurance [23]. The different study populations might cause this discrepancy, whereas we looked at adults with GERD, Bharmal et al. study looked at adults as a whole.

Regarding comorbidities, our study found that patients with heart disease, hypertension, diabetes, hyperlipidemia, asthma, COPD, arthritis, and cancer were negatively associated with HRQoL. Previous studies have described that individuals with chronic diseases experience lower HRQoL due to the chronic nature of the diseases and their management and that chronic diseases are related to mental health [24,25,26]. HRQoL has been investigated as a primary or secondary outcome in chronic illnesses research [26], as improving HRQoL is considered a significant outcome and a predictor of the therapeutic benefit of disease management. Health services could become more patient-centered with data on the influence of chronic conditions on HRQoL.

Factors positively associated with HRQoL in our study included marital status, employment, perceived general health, and physical activity; these findings are consistent with published studies. For example, employment has been positively associated with HRQoL in patients with multiple sclerosis [27] and liver transplant patients [28]. Physical activity has been reported to be positively associated with HRQoL by many published studies in the general adult population [29,30]. A systematic review of fourteen observational studies revealed a positive relationship between physical activity level and health-related quality of life [29].

### 4.1. Study Strengths and Limitations

This study used a nationally representative sample of US adult GERD sufferers, with a large sample size to examine how depression and anxiety related to their HRQoL, while previously published studies [14,15,16,17] were conducted in a single center and sample size of these studies ranged from 98 to 358 patients with GERD. We adjusted for several factors in our analysis, such as coexisting chronic conditions, physical activity, health insurance coverage, and sociodemographic characteristics. However, it’s crucial to consider a few limitations when evaluating the study’s results. Since GERD types and illness severity are unavailable in MEPS, they were not accounted for in the regression analysis. Additionally, the MEPS does not provide information relating to the degree of anxiety or depression, which may affect the HRQoL of individuals with GERD; as a result, the regression analysis did not consider this information. Furthermore, due to the cross-sectional nature of this study, establishing the causal relationship is challenging. Finally, the results cannot be extrapolated to older people because MEPS only provides information for adults.

### 4.2. Clinical and Practice Implications

The information reported in this study may be used by healthcare professionals and policymakers to improve healthcare planning and resource allocation to reduce the adverse consequences of anxiety and depression on individuals with GERD. Additionally, this study recommends that healthcare practitioners should regularly test for anxiety and depression and treat these conditions since early diagnosis and treatment can enhance HRQoL in individuals with GERD. This study has significance in public health as it promotes lifestyle modifications, such as exercise, which can greatly reduce the frequency and severity of mental health issues and GERD problems.

Gastroesophageal reflux disease (GERD) continues to be one of the most prevalent ailments that healthcare providers continue to treat. While proton pump inhibitors (PPIs) remain the preferred medical treatment for GERD, numerous publications have questioned their side effects, questioned the safety of long-term usage, and raised concerns about overprescribing PPIs [31]. The potential for treating GERD using pharmaceutical, dietary, surgical, and endoscopic methods is now better understood.

## 5. Conclusions

The results from this nationally representative sample shed insight on the relationships between low HRQoL and anxiety and depression among individuals with GERD. It also demonstrated the detrimental impacts of co-occurring chronic illnesses, low socioeconomic status, and the positive benefits of employment and exercise on HRQoL. This study emphasizes the clinical, political, and public health implications for better healthcare, allocation of resources, and promotion of lifestyle modifications to improve the HRQoL in patients with GERD.

## Figures and Tables

**Table 1 healthcare-11-02637-t001:** Characteristics of the Study Sample (n = 3068), N and Row % of Characteristics by GERD Group among Adults with GERD, Medical Expenditure Panel Survey 2017–2020.

	Total Sample		GERD Only		GERD & Depression		GERD & Anxiety		GERD & Dep & Anxiety		
	N	Wt.%	N	Wt.%	N	Wt.%	N	Wt.%	N	Wt.%	*p*-Value
All	3068	100.0	1898	63.0	405	13.3	394	13.2	371	10.5	
Age in years											
22–39	509	18.9	308	60.6	41	10.2	84	15.3	76	13.9	0.036
40–49	645	21.7	378	59.4	92	15.3	82	13.4	93	11.9	
50–64	1914	59.4	1212	65.1	272	13.6	228	12.4	202	8.9	
Gender											
Women	1820	56.4	1012	56.4	284	15.9	263	14.8	261	12.9	<0.0001
Men	1248	43.6	886	71.5	121	9.9	131	11.1	110	7.4	
Race/ethnicity											
White	2028	75.0	1189	60.8	281	14.0	296	14.3	262	10.9	0.048
African American	507	11.6	345	69.3	61	10.6	48	9.7	53	10.3	
Latino	360	8.2	244	71.9	35	8.4	34	9.2	47	10.5	
others	173	5.2	120	66.5	28	16.5	16	10.9	9	6.1	
Marital Status									371		
Married	1605	57.2	1088	67.7	180	11.9	196	13.0	141	7.4	<0.0001
Wid./Div./Sep.	873	25.1	479	56.2	143	16.8	121	13.3	130	13.8	
Never married	590	17.7	331	57.6	82	12.8	77	13.8	100	15.8	
Education Level											
LT HS	119	2.6	70	59.6	24	21.4	16	14.3	9	4.8	0.226
HS	315	8.2	186	58.7	39	12.5	44	14.3	46	14.5	
>HS	2612	88.6	1625	63.4	340	13.1	334	13.1	313	10.3	
Region											
Northeast	614	20.9	387	65.8	65	10.1	70	11.8	92	12.3	0.089
Mid-west	741	24.5	425	60.1	113	14.8	108	14.2	95	10.8	
South	1180	38.7	739	62.6	143	12.3	164	15.0	134	10.1	
West	533	15.9	347	64.8	84	17.6	52	8.8	50	8.8	
Employment											
Employed	1809	64.8	1253	69.2	185	10.9	209	12.0	162	7.8	<0.0001
Not employed	1259	35.2	645	51.6	220	17.7	185	15.3	209	15.4	
Poverty Status											
Poor	603	13.9	319	53.1	97	17.0	75	11.6	112	18.3	<0.0001
Near Poor	513	13.5	289	57.2	77	14.1	63	13.3	84	15.4	
Middle Income	810	26.5	479	60.1	104	12.6	133	16.4	94	11.0	
High Income	1142	46.1	811	69.4	127	12.4	123	11.8	81	6.4	
Health Insurance											
Private	1929	71.0	1306	67.6	232	12.4	222	12.2	169	7.8	<0.0001
Public	1068	27.4	543	50.6	169	16.3	164	15.7	192	17.4	
Uninsured	71	1.6	49	68.9	4	3.0	8	13.3	10	14.9	
Rx Insurance											
Rx insurance	1731	64.6	1188	68.3	205	12.6	196	11.9	142	7.2	<0.0001
No Rx insurance	1337	35.4	710	53.3	200	14.6	198	15.6	229	16.5	
General Health											
Excellent/very good	973	36.3	718	73.0	89	10.2	108	11.8	58	5.0	<0.0001
Good	1098	35.4	684	61.3	133	12.3	157	15.2	124	11.3	
Fair/poor	997	28.3	496	52.3	183	18.7	129	12.4	189	16.6	
Physical activity											
3/week	1211	41.4	841	71.4	123	9.2	141	11.7	106	7.7	<0.0001
No exercise	1842	57.9	1047	57.1	278	15.9	253	14.4	264	12.6	
Heart											
Yes	386	11.0	217	61.3	66	16.2	48	9.9	55	12.6	0.188
No	2682	89.0	1681	63.2	339	13.0	346	13.6	316	10.3	
Hypertension											
Yes	1498	45.3	850	58.7	231	15.2	223	14.9	194	11.2	0.009
No	1570	54.7	1048	66.6	174	11.8	171	11.7	177	10.0	
Diabetes											
Yes	670	18.6	353	53.2	129	18.7	85	13.8	103	14.3	<0.0001
No	2398	81.4	1545	65.2	276	12.1	309	13.0	268	9.6	
Hyperlipidemia											
Yes	1139	34.7	666	59.3	170	14.2	150	14.8	153	11.8	0.137
No	1929	65.3	1232	65.0	235	12.9	244	12.3	218	9.8	
Asthma											
Yes	593	17.2	286	51.1	84	14.3	108	17.2	115	17.4	<0.0001
No	2475	82.8	1612	65.5	321	13.1	286	12.3	256	9.1	
COPD											
Yes	331	9.6	151	46.5	53	15.4	56	16.5	71	21.6	<0.0001
No	2737	90.4	1747	64.7	352	13.1	338	12.8	300	9.3	
Arthritis											
Yes	658	18.8	349	52.9	108	17.2	86	13.1	115	16.9	<0.0001
No	2410	81.2	1549	65.3	297	12.4	308	13.2	256	9.0	
Osteoporosis											
Yes	43	1.5	14	25.8	12	36.2	11	29.4	6	8.6	<0.0001
No	3025	98.5	1884	63.6	393	13.0	383	12.9	365	10.5	
Cancer											
Yes	139	4.6	75	54.7	20	18.0	17	10.6	27	16.7	0.059
No	2929	95.4	1823	63.4	385	13.1	377	13.3	344	10.2	

COPD: Chronic Obstructive Pulmonary Disease; Dep: Depression; GERD: Gastroesophageal reflux disease; Rx: Medication; LT: less than; Wt.: weighted; Wid./Div./Sep.: widowed, divorced, and separated.

**Table 2 healthcare-11-02637-t002:** Health-related Quality of Life Scores by GERD Groups.

	Total Sample		GERD Only		GERD & Depression		GERD & Anxiety		GERD & Depression & Anxiety		
	Mean	SD	Mean	SE	Mean	SE	Mean	SE	Mean	SE	*p*-Value
HRQoL											
PCS	37.39	19.1	43.90	0.64	38.67	1.27	39.86	1.31	33.41	1.30	<0.0001
MCS	41.05	19.3	39.90	0.69	35.99	1.28	38.12	1.31	35.61	1.48	<0.0001

MCS: Mental Component Summary; PCS: Physical Component Summary; SE: Standard Error; SD: Standard Deviation.

**Table 3 healthcare-11-02637-t003:** Adjusted Multivariate Linear Regressions on Health-related Quality of Life among Adults with GERD, MEPS 2017–2020.

	Health-Related Quality of Life					
	PCS			MCS		
	Regression Coefficients	*p*-Value	Sig.	Regression Coefficients	*p*-Value	Sig.
GERD Groups						
GERD & depression	−0.654	<0.0001	***	−6.395	<0.0001	***
GERD & anxiety	1.215	<0.0001	***	−2.869	<0.0001	***
GERD & depression & anxiety	−0.368	<0.0001	***	−10.819	<0.0001	***
GERD only (Ref.)						
Age in years						
22–39	2.620	<0.0001	***	−1.636	<0.0001	***
40–49	0.177	<0.0001	***	−0.284	<0.0001	***
50–64 (Ref.)						
Gender						
Women	−0.561	<0.0001	***	−0.200	<0.0001	***
Men (Ref.)						
Race/ethnicity						
White	0.129	<0.0001	***	−1.441	<0.0001	***
Afr Am	0.898	<0.0001	***	0.034	<0.0001	***
Latino	1.858	<0.0001	***	1.322	<0.0001	***
others (Ref.)						
Marital Status						
Married	1.544	<0.0001	***	1.826	<0.0001	***
Widow/Sep/Div	1.055	<0.0001	***	−0.044	<0.0001	***
Never married (Ref.)						
Education Level						
>HS	1.424	<0.0001	***	2.640	<0.0001	***
HS	1.524	<0.0001	***	3.960	<0.0001	***
LT HS (Ref.)						
Region						
Northeast	1.532	<0.0001	***	1.595	<0.0001	***
Mid-west	0.320	<0.0001	***	0.854	<0.0001	***
South	−0.299	<0.0001	***	0.476	<0.0001	***
West (Ref.)						
Employment						
Employed	5.909	<0.0001	***	2.819	<0.0001	***
Not employed (Ref.)						
Poverty Status						
Poor	−2.230	<0.0001	***	−0.363	<0.0001	***
Near Poor	−1.599	<0.0001	***	0.402	<0.0001	***
Middle Income	−1.757	<0.0001	***	−0.394	<0.0001	***
High Income (Ref.)						
Health Insurance						
Private	−3.394	<0.0001	***	−0.878	<0.0001	***
Public	−1.785	<0.0001	***	−2.199	<0.0001	***
Uninsured (Ref.)						
Rx Insurance						
Rx insurance	2.420	<0.0001	***	1.219	<0.0001	***
No Rx insurance (Ref.)						
General Health						
Excellent/very good	10.457	<0.0001	***	5.627	<0.0001	***
Good	6.218	<0.0001	***	3.958	<0.0001	***
Fair/poor (Ref.)						
Physical Activity						
3/week	1.846	<0.0001	***	0.481	<0.0001	***
No exercise (Ref.)						
Heart						
Yes	−2.853	<0.0001	***	−0.277	<0.0001	***
Hypertension						
Yes	−1.480	<0.0001	***	0.355	<0.0001	***
Diabetes						
Yes	−1.850	<0.0001	***	1.129	<0.0001	***
Hyperlipidemia						
Yes	−0.528	<0.0001	***	−0.761	<0.0001	***
Asthma						
Yes	−1.172	<0.0001	***	0.299	<0.0001	***
COPD						
Yes	−1.357	<0.0001	***	0.485	<0.0001	***
Arthritis						
Yes	−3.572	<0.0001	***	0.225	<0.0001	***
Osteoporosis						
Yes	6.278	<0.0001	***	−1.906	<0.0001	***
Cancer						
Yes	−1.847	<0.0001	***	−2.693	<0.0001	***

Asterisks denote statistical significance in parameter estimates from multivariate linear regressions on health-related quality of life. *** *p* <0.001. GERD: Gastroesophageal reflux disease; HS: High School; LT: less than; MCS: Mental Component Summary; PCS: Physical Component Summary; Rx: Medication; Ref: reference group; Sig: Significance. Wid./Div./Sep.: widowed, divorced, and separated.

## Data Availability

Researchers can access the publicly accessible dataset used in this study from the MEPS database at this URL: https://meps.ahrq.gov/data_stats/download_data_files.jsp (accessed on 8 June 2023).

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
