# Peer review of "Anxiety and Depression and Health-Related Quality of Life in Adults with Gastroesophageal Reflux Disease: A Population-Based Study"

_healthcare, 2023, doi:10.3390/healthcare11192637_

Round 1
Reviewer 1 Report
Well written article. Interesting study design. However, in my opinion the study will get higher scientific soundness if the therapeutic management is descriped as well ( before the conclusion) .
Author Response
Comment # 1: Well written article. Interesting study design. However, in my opinion the study will get higher scientific soundness if the therapeutic management is described as well (before the conclusion).
Response: Thanks for your valuable comment; we have now described the therapeutic management in the discussion (Page 9, lines 253-258).
Reviewer 2 Report
In title location is needed (also in abstract, aim section)
In abstract, Methods, study type, location, time and sample size and sampling method is needed.
In abstract Results, a description of basic and demographic variable is needed.
This 13% had anxiety, 13% had depression, and 10% had both is strong, in general population prevalence is higher than 13%.
In introduction, details such as HR=2.99, 95% CI=2.12-4.22) or (β = -0.39; p 0.01) not needed. Also in discussion.
Line 65 -77 hasn’t reference, id possible be shorter.
The validity, reliability and scoring of the questionnaire should be stated clearly.
Sampling should be stated,
Ethical considerations are not seen in the method.
Table 2 is unclear and confusing.
Which method of regression was used? Backward?
In table 3 GERD Groups why included in the model?
13% had anxiety, 13% had depression, and 10% had both. Not appear in the main text. Add it .
Discussion needs revision. It’s short, it should be more complete.
Author Response
Comment # 1: Title and Abstract
- In title location is needed (also in abstract, aim section)
- In abstract, Methods, study type, location, time and sample size and sampling method is needed.
- In abstract Results, a description of basic and demographic variable is needed.
Response: Thanks for your valuable comments
- We have now added the location to the study title.
- We have added in the Abstract methods section: Study design, location, the time, sampling method [1,2], and sample size.
- We have added in the abstract Results section, a description of demographic variables.
Reference:
- Agency for Healthcare Research and Quality. 2021 National Healthcare Quality and Disparities Report Detailed Methods for the Medical Expenditure Panel Survey. 2021 [cited 2023]; Available from: https://www.ahrq.gov/sites/default/files/wysiwyg/research/findings/nhqrdr/2021qdr-mepsmethods.pdf#:~:text=MEPS%20uses%20an%20overlapping%20panel%20design%20in%20which,2%20calendar%20years%20are%20collected%20from%20each%20household.
- Agency for Healthcare Research and Quality. https://meps.ahrq.gov/data_files/publications/mr33/mr33.shtml
Comment # 2: Introduction
- In introduction, details such as HR=2.99, 95% CI=2.12-4.22) or (β = -0.39; p 0.01) not needed. Also in discussion.
- Line 65 -77 hasn’t reference, id possible be shorter.
Response: Thank you for pointing this out.
- We have removed the HR and β from the introduction as well as the discussion.
- We have added the references, and shorten the paragraph.
Comment # 3: Methods
- The validity, reliability and scoring of the questionnaire should be stated clearly.
- Sampling should be stated,
- Ethical considerations are not seen in the method.
- Table 2 is unclear and confusing.
- Which method of regression was used? Backward?
Response: Thank you for pointing this out;
- We have added the following sentence with reference to the methods section:
- The MEPS's Mental Component Summary Scores (MCS) and Physical Component Summary Scores (PCS) were both found to have high internal consistency reliability, adequate reliability, and validity, and should be appropriate for use in a variety of database-based projects in groups with and without cognitive limitations [3] (Page 3, lines 103-106).
- Sampling methods of MEPS database was added (Page 2, lines 86-88).
- This is a publically available data with not patient identifier, therefore no Ethical considerations is required (Page 9, lines 273-275).
- Table 2 display the mean scores of the Mental Component Summary Scores (MCS) and Physical Component Summary Scores (PCS), (bivariate analysis using ANOVA) which is usually conducted before the multivariate analysis in Table 3 (with adjustment for other confounders).
- We have used forward linear regression analysis.
Reference:
- Cheak-Zamora NC, Wyrwich KW, McBride TD. Reliability and validity of the SF-12v2 in the medical expenditure panel survey. Qual Life Res. 2009;18(6):727-35.
Comment # 4: Results
- In table 3 GERD Groups why included in the model?
- 13% had anxiety, 13% had depression, and 10% had both. Not appear in the main text. Add it .
Response: Thank you for pointing this out;
- The primary goal of this research is to determine, the link between anxiety, depression, and HRQoL in individuals with GERD, after taking into account a number of confounders. Therefore, in the multivariate forward linear regression model (Table 3), we have added GERD groups as a factor and adjusted for other confounders.
- We have added the prevalence of anxiety, depression, and both in the main text (Page 3, lines 178-179).
Comment # 5: Discussion
Discussion needs revision. It’s short, it should be more complete.
Response: Thank you for this suggestion; we have now expanded the discussion part.
Reviewer 3 Report
I read the article very carefully. The paper focused on a very simple topic, and although it might have had some interest for the scientific community I think many aspects should have been explored in depth.
I think there is a lack of a theoretical framework on the topic
methodologically, it is unclear how anxiety and depression was measured in patients
from a methodological point of view, there is a lack of a control group that has no diagnosis of GERD
in light of these general considerations, I believe the paper cannot be published even if there was a major review
Author Response
Comments:
- I read the article very carefully. The paper focused on a very simple topic, and although it might have had some interest for the scientific community I think many aspects should have been explored in depth. I think there is a lack of a theoretical framework on the topic methodologically,
- it is unclear how anxiety and depression was measured in patients from a methodological point of view,
- there is a lack of a control group that has no diagnosis of GERD in light of these general considerations, I believe the paper cannot be published even if there was a major review
Response: Thanks for your comments; below is our response to each comment:
- In depth understanding of the topic and use of theoretical framework in future research is one of our research implications.
- MEPS data is a nationally representative data in the United States. The medical conditions (including anxiety and depression) reported by the Household Component respondent were recorded by the interviewer as verbatim text and then were coded to ICD-10-CM codes (ICD10CDX) by professional coders (link: https://meps.ahrq.gov/data_stats/download_data/pufs/h214/h214doc.shtml#Diagnosis).
- The use of control group was not necessary since we are not comparing GERD group with the general population, instead we are comparing GERD group with or without mental comorbidity.
I hope I have addressed your concern in this response.
Reviewer 4 Report
This is a well-written manuscript that explores the influence of gastroesophageal reflux disease on mental health and identifies detrimental impacts of co-occurring chronic illnesses, low socioeconomic status, as well as the positive benefits of employment and exercise on quality of life. The manuscript has valuable contributions to add to the psychiatric literature.
General concern:
1. The sample population was drawn from the Medical Expenditure Panel Survey (MEPS) but only includes those that completed all SF-12 surveys for the time-period studied. Consequently, it is possible that a significant proportion of the MEPS population was excluded from the analysis. Knowing the characteristics of the excluded population would assist the reader in understanding how to interpret the findings.
Specific concern:
1. In the Discussion section, lines 194-199, it is mentioned that the current manuscript has findings that are discrepant to a prior publication. It would be helpful to offer some ideas on why this discrepancy occurs.
Author Response
This is a well-written manuscript that explores the influence of gastroesophageal reflux disease on mental health and identifies detrimental impacts of co-occurring chronic illnesses, low socioeconomic status, as well as the positive benefits of employment and exercise on quality of life. The manuscript has valuable contributions to add to the psychiatric literature.
Comment # 1: General concern:
- The sample population was drawn from the Medical Expenditure Panel Survey (MEPS) but only includes those that completed all SF-12 surveys for the time-period studied. Consequently, it is possible that a significant proportion of the MEPS population was excluded from the analysis. Knowing the characteristics of the excluded population would assist the reader in understanding how to interpret the findings.
Response: Thank you for your thoughtful comment. In order to conduct the analysis and compare the health-related quality of life between GERD groups with and without depression and anxiety, we have only included those who completed the SF-12 survey. Unfortunately, we haven't examined the characteristics of the population that was excluded.
Comment # 2: Specific concern:
- In the Discussion section, lines 194-199, it is mentioned that the current manuscript has findings that are discrepant to a prior publication. It would be helpful to offer some ideas on why this discrepancy occurs.
Response: Thank you for pointing this out, we have now explained why this discrepancy could exist between our study and the existing published study by Bharmal et al. (Page 8, lines 251-252).
Round 2
Reviewer 3 Report
From my personal point of view, The paper now is ready to be published. Thanks a lot